# Audio Legends: Investigating Sonic Interaction in an Augmented Reality Audio Game

**Emmanouel Rovithis** *,†,‡ 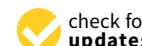, **Nikolaos Moustakas** †,‡, **Andreas Floros** †,‡ **and Kostas Vogklis** †,‡

Department of Audiovisual Arts, Ionian University, GR49100 Corfu, Greece; a11mous@ionio.gr (N.M.); floros@ionio.gr (A.F.); voglis@cs.uoi.gr (K.V.)
* Correspondence: emrovithis@gmail.com; Tel.: +30-26610-87725
† Current address: Audiovisual Signal Processing Lab., Plateia Tsirigoti 7, GR49100 Corfu, Greece.
‡ These authors contributed equally to this work.

**Abstract:** Augmented Reality Audio Games (ARAG) enrich the physical world with virtual sounds to express their content and mechanics. Existing ARAG implementations have focused on exploring the surroundings and navigating to virtual sound sources as the main mode of interaction. This paper suggests that gestural activity with a handheld device can realize complex modes of sonic interaction in the augmented environment, resulting in an enhanced immersive game experience. The ARAG "Audio Legends" was designed and tested to evaluate the usability and immersion of a system featuring an exploration phase based on auditory navigation, as well as an action phase, in which players aim at virtual sonic targets and wave the device to hit them or hold the device to block them. The results of the experiment provide evidence that players are easily accustomed to auditory navigation and that gestural sonic interaction is perceived as difficult, yet this does not affect negatively the system's usability and players' immersion. Findings also include indications that elements, such as sound design, the synchronization of sound and gesture, the fidelity of audio augmentation, and environmental conditions, also affect significantly the game experience, whereas background factors, such as age, sex, and game or music experience, do not have any critical impact.

**Keywords:** audio games; audio augmented reality; audio augmented games; augmented reality audio games

## 1. Introduction

Audio Games (AG) are electronic games that employ sound as the main carrier of information, thus requiring players to rely on the auditory channel, in order to perceive and interact with the game world. Augmented Reality (AR) on the other hand enhances the real world with computer generated objects in real time, thus granting users the ability to perceive and interact with virtual information otherwise not directly detectable. Recently, both fields have benefited from the rapid progress of mobile technology, which allowed AG and AR applications to become portable and available to the broader public through smartphones and tablets. However, their combination remains rather under-researched; existing Augmented Reality Audio (ARA) systems have been developed mainly for navigational aid and for non-linear exploration of closed and open spaces, whereas the very few implementations of Augmented Reality Audio Games (ARAG) have built their mechanics on moving to audio sources and pressing a virtual button. More complex interaction modes, such as using the device to perform gestures, are yet to be explored.

Based on their previous work on AG and ARAG design [1–4], the authors argued that ARAG can utilize gestural activity to express challenging tasks that will enhance players' active participation

and immersion in the game environment. To that scope, an ARAG entitled "Audio Legends" was developed to include both navigational and gestural sonic interaction. The results obtained through testing sessions were very positive, suggesting that complex sonic interaction was feasible and well accepted in terms of usability and immersion.

The paper is structured as follows. Section 2 deals with the evolution and characteristics of AG and ARA systems including existing ARAG approaches. Section 3 presents the implementation process of Audio Legends in terms of scenario, mechanics, programming, and sound design, while Section 4 describes the experiment performed and discusses the results. Finally, Section 5 provides a summary of the conclusions and outlines potential future research considerations.

## 2. Towards Augmented Reality Audio Games

### 2.1. The Evolution and Characteristics of Audio Games

AG designers defy the conventional trend in the video game industry to strive for increasingly realistic and complex graphics, and instead pursue various data sonification and sonic interaction design techniques, in order to express the game's content and mechanics in ways matching the ones facilitated by their audiovisual counterparts. The implementation of such techniques not only within the context of a game environment, but also in a wider range of fields, such as educational or therapeutic applications, is widely supported in the literature to have a multifaceted positive impact on the user. The manipulation of aural information has been reported as an effective way to interact with a complex auditory space [5]. This notion is buttressed by later research suggesting that spatialized sound facilitates the accurate reconstruction of space [6], whereas audio-based interfaces facilitate users in correctly navigating virtual worlds and transferring the acquired spatial information to real-life situations [7]. Other research works focus on the natural properties of sound suggesting that multi-layered sonic interaction promotes users to gain and retain attention on the appropriate information and subsequently relate that information to a larger system of conceptual knowledge [8], as well as exhibiting great potential in helping users become more proficient at fine movements and the complicated manipulation of tools [9]. Interacting with sonic stimuli has also been proven to assist in increasing memory and concentration [10], as well as in triggering emotional responses in ways not possible through visual means [11]. The exclusion of the visual in favor of the auditory channel results in more subjective interpretation and thus boosts players' fantasy [12]. Furthermore, turning graphics off to rely on purely audio stimuli can result in increased immersion in the virtual environment: one such research compared playing AG with visual aids turned on and off [13], while another suggested that players gain physical freedom in their spatial behavior within a 360° field of actions when no longer confined by looking at a screen [14].

Another important factor for the evolution of AGs is the technological context used for their realization. So far, the genre has shown remarkable adaptability in the ways it has addressed the audience. Even though AG titles are by far fewer than electronic video game releases, they have adopted a similar variety of technologies; they have been released as arcade cabinets, handheld devices, console interfaces, personal computer stand-alones, online environments, and mobile applications. The majority of AGs may have been designed to run on a personal computer, but quite a few implementations on other platforms have met with great success. "Simon", one of the earliest prototypes released in 1978 as a handheld memory game, in which players have to repeat a growing sequence of tones, became one of the top-selling toys and a pop culture symbol of its time [15]. "Bop It", another handheld device with parts such as a button, a lever, and a handle that players interact with as instructed, was welcomed to the market in 1996 [16]. In the meantime, text-to-speech software had made a variety of narration-based works, from interactive books to role-playing and adventure games, accessible in audio-only forms. In the console era, some very popular rhythm-action games emerged that used external controllers to add kinesthetic features to their game play, facilitating players to synchronize their actions and movement with the music. "Dance Dance Revolution"

released in 1998 [17] and "Guitar Hero" released in 2005 [18] are characteristic examples of that trend. "Soundbeam", a music creation environment with incorporated haptic devices for gestural control, has been found to have therapeutic impact on the emotional and communicational skills of children with multiple disabilities [19]. "Soundvoyager", a collection of audio puzzles released in 2006 for the Game Boy Advance [20], signifies that handheld mini-consoles are also fertile ground for audio mechanics, whereas the Tangible Audio Game Development Kit demonstrates the potential of AG prototyping through the manipulation of haptic artifacts [21].

Connectivity facilitated by the Internet has resulted in multiplayer interaction modes like in the case of Memor-I Studio, an open-source platform allowing the creation of AG with online player vs. player capability [22]. Particularly in the last decade, mobile smartphones, a massively accepted technological medium, have become the incubator of various successful AGs, such as "Papa Sangre", a horror-themed adventure game based on binaural audio [23], "Audio Game Hub", a collection of accessible arcade games [24], and "Bloom", an interactive environment for artistic musical experimentations [25]. The fact that audio interaction can be even totally screen independent combined with the growing processing power and haptic capabilities of current-generation smartphones, which include touchscreen, GPS, compass, accelerometer, gyroscope, and other sensor modules, foreshadows a promising future for AG design on mobile platforms.

Throughout its history, the AG genre has not ceased to benefit from technological advances. Powerful companies in the toy/game industry, as well as smaller developers, have often succeeded in addressing AG to the broad public through different technological media. The main audience of AG remains the visually impaired community; however, the game market is gradually becoming more aware of the genre's potential. On the other hand, academic research has been increasingly dealing with the issues of data sonification and sonic interaction design, focusing not only on accessibility, but also on factors that drive the immersion in the audio-only experience. The AG genre has progressed from an "exotic" medium to a systematically studied field, as in projects like the Online Audio Game Editor, which aspires to build a community of creators investigating the AG design process [21]. In the context of the above, the authors suggest that the conditions are met for merging AG with Augmented Reality (AR) towards the development of Augmented Reality Audio Games (ARAG). Fusing these technologies will be mutually beneficial as it will enrich AG mechanics with kinetic and gestural behavior, as well as grant AR mechanics the merits of audio interaction.

## 2.2. Approaches to Audio Augmentation

Augmented Reality Audio (ARA) is a type of AR, in which the virtual component that enriches the real world consists of audio information. Users of an ARA system perceive their surroundings in a "pseudoacoustic" form as the natural acoustic environment is mixed with artificially created sounds. A typical implementation to achieve this involves a special pair of headphones with integrated microphones, through which the real sound is captured, mixed with the virtual components and returned to the user. In that process, the fundamental principles of AR systems must be adhered to: the system should combine the real with the virtual, be interactive in real time, and registered in three dimensions [26].

ARA has been implemented in diverse ways in its recent attempt to make its own path against the conventional visual AR. Outdoor playgrounds have been augmented by attaching sonic interaction mechanisms to physical objects [27]. QR codes carrying audio information have been assigned to locations and museum exhibits. Mobile platform audio applications, such as RjDj, transform the acoustic environment in real time and make it adapt to the music played [28]. However, this research deals with the most common form of ARA, which is based on the principle of attaching virtual sounds to locations for exploration and interaction. Such setups are often composed of two concentric levels of audio feedback, one in wide proximity providing cues that guide users towards it and the other in a narrower activation zone allowing further interaction [29]. Mobility is facilitated by sensor mechanisms that locate users' position and track their movement in indoor or outdoor environments [30],

whereas three-dimensional space is commonly measured as roll, elevation, and azimuth, to allow for finer positioning of sonic objects and tracking of complex user behavior [31].

Sonic interaction within such AR environments has been found to have a multifaceted positive impact on the user. Research results have shown that sonic stimuli are equally effective as visual maps in guiding users to points of interest [32]. Audio interaction can help users navigate to locations without distracting them from their environment [33] and regardless of their musical training [34]. Another study suggests that the visual display bears some weaknesses that the audio modality can overcome, such as limited screen space, overload of information, vulnerability to sunlight, and the necessity for constant attention [35]. Furthermore, many experiments argue that augmentation through data sonification and sonic interaction design techniques can significantly enhance players' immersion level in the augmented environment [29,36] and increase their emotional engagement with the virtual world [37,38]. It has also been observed that in some cases, users could not distinguish the real from the virtual component [39]. Nevertheless, the notion that ARA is still under-researched and needs to be systematically studied is widely supported in the literature. Sound design is only given a marginal role in the development of location based games [36], and there is a lack of design guidelines for non-visual display modes of AR applications [35]. Sound appears to be an unfamiliar game mechanic to sighted players [40]. The auditory culture of space itself should be considered more thoroughly [27,41]. Last, the technical requirements for an efficient ARA experience should be further investigated, including spatialization techniques, headset usage, GPS and other sensors' accuracy, and computation power [29,38,42–44].

ARA systems of the type that is investigated by this work have been developed in three major directions in terms of the scope underlying their audio interaction mechanisms: "navigation", "exploration", and "entertainment". This categorization aims more to help in the organization of the projects investigated rather than to serve as an absolute distinction between them. Their difference lies in the main purpose for their design. Projects ranked by the authors in the navigation category are essentially guidance systems to specific destinations, whereas projects in the exploration category provide information about a multitude of places for the user to visit at will. Last, projects in the entertainment category employ gaming features, such as objectives, rules, and score, in the design of user experience. Nevertheless, it must be noted that some overlap between the three categories surely exists.

Representing the first category, "Audio GPS" employs non-speech, spatial audio for eye-and-hand-free navigation towards a destination [34], whereas "GpsTunes" guides users to the desired location by modifying the volume and panning of continuous music feedback and featuring a browsing mode, in which users point their device around to discover more target destinations [45]. Spatialized contextualization of the user's own audio content for auditory navigation has been also applied in the "Melodious Walkabout" project [46]. Implementations in the exploration category are not so strictly bound to guiding to a specific destination; instead, they facilitate the non-linear exploration of multiple points of interest. One major field of application for such systems is museums. The "LISTEN" project aimed at facilitating museum tours through virtual acoustic landmarks [47], whereas "ec(h)o" tracked the visitor's position in the exhibition space to play back the respective soundscape and used a vision system to allow for interaction with audio cues through gestures [48]. Outside the museum space, ARA exploration applications can be applied on specific sites or even whole cities [49]. The "Roaring Navigator" features multiple user access to sound clips conveying the position and relative information of animals in a zoo, complemented by a speech recognition module, which allows for verbal use of the system [33], whereas in [29], an experiment was carried out to measure user behavior in an audio augmented park. In the "Riot! 1831" installation at the Queens Square in Bristol, visitors could trigger sound files based on real events that had occurred there by walking to various spots [37]. In the "Soundcrumbs" project, users could attach different sounds to locations like traces on their trail [50]. In a similar context, "Tactical Sound Garden" is a WiFi based open source platform that enables users to attach sounds to locations for public use [41], whereas in

"Audio Bubbles", virtual audio spheres are placed on physical landmarks to help tourists in their way-finding [32]. Last in this category, the project discussed in [43] complements the aforementioned exploration mechanism with tactile interaction, in which users discover nearby locations by dragging their finger to cross sound nodes.

The entertainment category has its roots in GPS based art exhibitions that used the movement of the participants for real time control of a music composition [30]. Projects that add game rules and objectives to this principle realize the ARAG concept. One such example is the mobile game "Zombies Run!", in which players run away from approaching zombies detectable through the sound they make [51]. Further examples based on simple movement in the augmented space are "SoundPacman", in which players follow audio cues to collect virtual cookies positioned across the streets [36], and "Guided by Voices", where players walk through locations marked by radio frequency beacons attached to physical props and thus progress in a medieval fantasy narrative by triggering the respective sound events [52]. Other projects have added more complex modes of audio interaction to just navigating to the sound. In "Viking Ghost Hunt", once players enter the closest proximity zone of a virtual sound, they use the game's visual interface to record the audio sample [38]. In "Eidola", players have to locate the virtual sounds and then press a button on the display once standing at their position [1], whereas in the multi-player version of the same game, the movement of the virtual components is controlled by the movement of other players in a different room [53]. In "The Songs of North", players press a virtual drum on the display to interact with the game world [40]. Finally, in "SoundPark", players co-operate under different roles to accomplish the game's objective: one member of the team discovers the location of a virtual sound; another aims to collect it; and a third one organizes the gathered sounds in the correct order [44].

The aforementioned projects are to the authors' knowledge the documented attempts in the ARA field that utilize sonic interaction in the context of a game. However, few of them go beyond the point of building their mechanics entirely upon simple physical movement in the augmented space to feature modes such as scanning and tapping. The authors suggest that more complex gestural activity with the handheld device can enhance ARAG sonic interaction and contribute to a challenging and immersive game experience. This hypothesis is made by the authors based on pieces of research that stress the need to investigate gesture based interaction. In [54], it was argued that user input transcending simple key presses or mouse clicks can rely on general sensorimotor levels of knowledge and thus realize intuitive user interfaces. Since humans use their hands as the main instrument for the manipulation of objects in real life, free hand interaction should be employed for the design of AR interfaces that will be accepted by end users [55]. Gesture interaction for handheld AR in particular directly utilizes the user's natural behavior; however most research related to the matter has focused on investigating the movement of fingers rather than of hands [56]. In the process of designing Audio Legends, the adoption most of the existing gesture based interaction modes and the inclusion of new ones were taken into consideration. Thus, this investigation aims on the one hand to confirm the existing results on established ARA practices, such as scanning the acoustic environment and navigating to a virtual sound, and on the other to measure the impact of gestural activity, such as waving the device to hit virtual objects or hold it to block them.

## 3. Audio Legends' Implementation

Audio Legends is an ARAG designed to investigate sonic interaction as a means not only to navigate within the augmented environment, but also to perform gestures required to achieve the game's objectives. To investigate the implications of such mechanics regarding the usability of the system and the immersion in the game world, it was designed to include different long and close range tasks. The scenario to host them was designed by drawing on folktales of Corfu island in Greece. More specifically, the prototype was based on the local myths regarding the protector of the island, Saint Spyridon, who is believed to have saved the island on four different occasions:

- In the 16<sup>th</sup> Century, he ended famine by guiding Italian ships carrying wheat out of a storm to the starving Corfiots.
- Twice in the 17<sup>th</sup> Century, he fought the plague off the island by chasing and beating with a cross the (described as) half-old woman and half-beast disease.
- In the 18<sup>th</sup> Century, he defended the island against the naval siege of the Turks by destroying most of their fleet with a storm.

In relation to these achievements, three Game Modes (GM) were designed:

- "The Famine", in which players have to locate a virtual Italian sailor and then guide him to the finish point.
- "The Plague", in which players have to track down and then defeat in combat a virtual monster.
- "The Siege", in which players have to find the bombardment and shield themselves from incoming virtual cannonballs.

*3.1. Mechanics' Design*

All GM consist of two interaction phases: exploration and action. The former is related to locating the virtual sound position and navigating to its proximity, whereas the latter is initiated, when players enter a 5 m radius around the sound position, and focuses on gestural behavior in response to virtual sonic stimuli. In both phases, no visual contact whatsoever is required with the game's interface, which is hidden behind a blank screen.

In the exploration-phase, standard ARA techniques regarding amplitude and panning facilitate the localization of the virtual sound. The field of the game is divided into concentric proximity zones around the emitting source. The position of the player is GPS tracked and assigned to the respective zone. Amplitude is then scaled logarithmically: the closer one gets to the source, the louder it becomes. Three-dimensional positioning is dynamically modified in relation to the direction the device is held at. The only difference between the three GM is that in Famine and Plague, the sound is only heard when the tablet is facing it, whereas in Siege, the sound is heard independently of the device's orientation. Thus, in Famine and Plague, players are required to use the device as an instrument to scan their surroundings.

During the action-phase, the three GM differentiate themselves considerably from one another. In Famine, once players enter the sound's closest range, it becomes attached to them so that they can lead it to the required destination. Every 20 to 30 s, the sound stops following them and moves to the closest one of the pre-distributed positions along the possible tracks. Players must then recollect the sound and repeat that process until the completion of their objective. In the action-phase of the Plague, the sound moves between random positions within a 150° azimuth and 60° elevation conical field in front of the player, who needs to aim the device at the sound's direction and perform an abrupt front and back gesture like hitting a nail with a hammer. Seven successful hits complete the mode's objective. Finally, in the action-phase of the Siege, the sound originating from random positions within the same frontal cone moves towards the player following a curved path. Players must estimate the sound's trajectory and hold the device like a shield to block the anticipated impact. If they do not succeed, the sound will pass right through them. Five successful blocks win the GM. Figure 1 demonstrates the three different gestures applied using the handheld device in the action-phase of the Famine, Plague, and Siege, respectively.

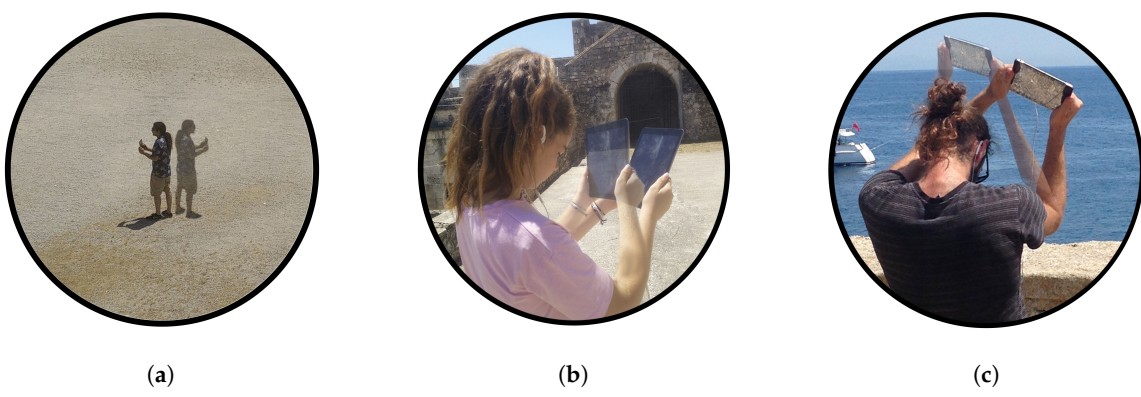

|(**a**)|(**b**)|(**c**)|

**Figure 1.** The applied gestural interactions. (**a**) Scanning. (**b**) Hitting. (**c**) Blocking.

### 3.2. Technical Implementation

Audio Legends was designed for an open area of 120 × 46 m. Outdoor virtual stimuli positioning was performed with the minimum attainable consumer GPS accuracy of 10 m. The game was developed on Xcode 10.2 using Swift 5. A major design principle followed was the compatibility with as many iOS devices as possible. For this reason, the bare basics Swift libraries were employed: core location (for GPS location events), MapBox (a library to display position on map), core motion (for device specific motions), and SceneKit (for rendering 3D audio). The end-user equipment consisted of a binaural audio recording headset (Sennheiser Ambeo Smart Headset) and an iPad Air (early 2014 model). In the exploration-phase, core location was combined with MapBox to create a mapping between the actual geographical coordinates of the user and the virtual 3D coordinates of SceneKit. The distance and the orientation in the virtual world (SceneKit) were used to render spatial audio. In the action-phase, the core motion library was used to get the orientation of the device, and SceneKit was employed to render moving audio targets in 3D. Geolocation was disabled during this phase, and moving objects were always oriented in front of the player regardless of the orientation, in which they entered the scene. The iOS libraries involved in each phase are shown in Figure 2.

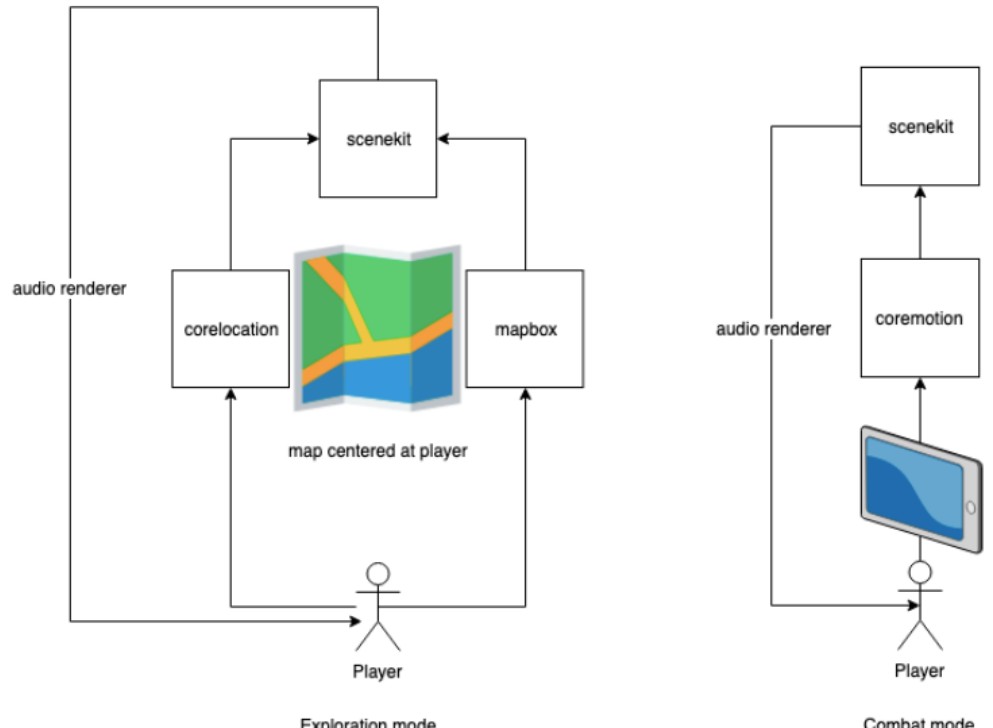

**Figure 2.** The iOS libraries involved in each phase.

One major challenge was that SceneKit audio rendering does not allow direct control of the audio level. Once a virtual object is set with the appropriate sound, the engine performs all necessary calculations for sound spatialization. In order to achieve the scanning feature and since the sound level could not be directly modified, the position of the virtual object was changed through appropriate movements along the z-axis. When the device was aiming directly at the target, the virtual object was positioned at the zero level, otherwise it was positioned appropriately high in the z-axis. The second challenge encountered had to do with the sound dynamics as one moves closer to the virtual audio source. The dynamic amplification by SceneKit was not enough to produce a perceptually accurate proximity effect. Thus, all virtual object positions were scaled down by a constant factor and brought closer to the player's ears. The scaling factor was applied gradually to achieve the desired proximity effect. It was an inverse function of the distance between the player and the target. A third challenge involved the mechanics during the action-phase. In order to simplify the interaction, the scene was rendered directly in front of the player regardless of the magnetic orientation of the device. Thus, a comfortable front field of view was achieved that allowed tilting and panning, under the restriction that the players could not perform a 180° turn around themselves.

*3.3. Sound Design*

The sonic content of Audio Legends consisted of recorded or electronically created samples, which were processed using an audio editor and sequencer and various sound design software modules. All samples intended for spatial positioning were monophonic. The only stereophonic sounds were the sound of rain adding an atmospheric effect to the action-phase of the Siege and the high pitched sound of a bell indicating the transition from exploration to action. The latter sound and the mixed sound of an ambient pad and a choir marking the successful completion of each GM were the only common sounds present in all GM.

In Famine, the player needs to navigate to the sound of a voice speaking in Greek with an Italian accent. The words of the "Italian" sailor were: "We all heard a voice. Where is the old man? The one that saved us. We bring wheat to the people of the island!". In the action-phase, the sound attached to the player's movement consisted of footsteps. When the footsteps stopped, the same voice waited to be relocated, speaking this time: "I am so tired. Are we there yet? I have so many questions. I'll just rest here for a while.". In Plague, the sound of the monster was created by mixing various mechanical sounds played back in a very slow speed for a spooky crawling effect with a periodic filtered noise gesture resembling a breath. During the action-phase, players received feedback on their hits through a woman's voice, who either laughed, when they missed, or groaned in pain, when they succeeded. In Siege, the sound to be located in the exploration-phase consisted of an arrhythmic bombing sequence. In the action-phase, the cannon ball sound to be blocked consisted of three parts: a shot in the beginning, then a six-second burst of filtered noise with rising pitch, and an explosion appearing to resonate behind the player's position. Successful blocks were signified by the sound of thunder.

## 4. The Evaluation Process of Audio Legends

The evaluation of the game dealt with two specific issues: usability and immersion in the game world. Moreover, the collection of demographic and technographic data allowed for the investigation of background factors that may influence the audience's receptivity to ARAG systems. Thus, the Research Questions (RQ) were formulated as follows:

RQ1: Is the utilization of complex navigational and gestural sonic interaction in an augmented reality environment (a) technically feasible and (b) capable of facilitating an easy to use and satisfactory game experience?

RQ2: Can sonic interaction and sound design techniques express an immersive augmented game world that motivates players to explore and interact?

RQ3: Are factors, such as age, sex, and familiarity with technological and musical elements, critical to the understanding and enjoyment of ARAG?

*4.1. Assessment Methodology*

To address the above research questions, three end-user questionnaires were designed. The first one collected demographic data including users' age, sex, and occupation, as well as technographic information that helped to understand users' technological fluency and experience with similar systems. The second questionnaire referred to a specific game mode (if subjects participated in two or all three game modes, they would fill in two or three of these respectively), whereas the third one dealt with the overall game experience. Both of the latter featured statements that were evaluated through a one to five Likert scale ranging from "strongly disagree" to "strongly agree". The statements were formulated in a mixed positive and negative way to protect from wild-card guessing. When analyzing the results, the evaluations of the negative statements were inverted to match the scaling of the positive ones.

The statements with respect to the scopes of the investigation were distributed as follows:

- Game Mode Specific Questionnaire (GMSQ)

  - Technical aspect (seven statements)

    * It was difficult to complete the game
    * Audio action and interaction through the device were successfully synced
    * It was difficult to locate the sound during the first phase of the game
    * It was difficult to interact with the sound during the second phase of the game
    * The sounds of the game didn't have a clear position in space
    * The sounds of the game didn't have a clear movement in space
    * The real and digital sound components were mixed with fidelity

  - Immersion aspect (five statements)

    * I liked the game
    * The sounds of the game helped me concentrate on the gaming process
    * The sounds of the game had no emotional impact on me
    * The game kept my interest till the end
    * I found the sound design of the game satisfying

- Overall Experience Questionnaire (OEQ)

  - Technical aspect (two statements)

    * The prolonged use of headphones was tiring/annoying
    * The prolonged use of the tablet was tiring/annoying

  - Immersion aspect (four statements)

    * I found the game experience (1 very weak – 5 very strong)
    * During the game I felt that my acoustic ability was enhanced
    * Participating in the game process enhanced my understanding of information related to the cultural heritage of Corfu
    * I would play again an Augmented Reality Audio Game

Some of the statements were derived from research on AG immersion [57], and the rest were designed by the authors. The use of the Likert scale facilitated the quantification of qualitative data for further analysis. Additionally, an open-ended question was included for any comments the participants felt they should submit. Apart from the subjective data, objective data measured on user performance included the successful or not completion of each GM, the respective time taken, and the reason for any help asked for in the process.

*4.2. Conducting the Experiment*

The experiment was organized at the Old Fortress of Corfu (Figure 3), a highly attractive location that provides a large and quiet open space with a view to the sea along the southern side. An ancient temple marks the eastern boundaries of the site.

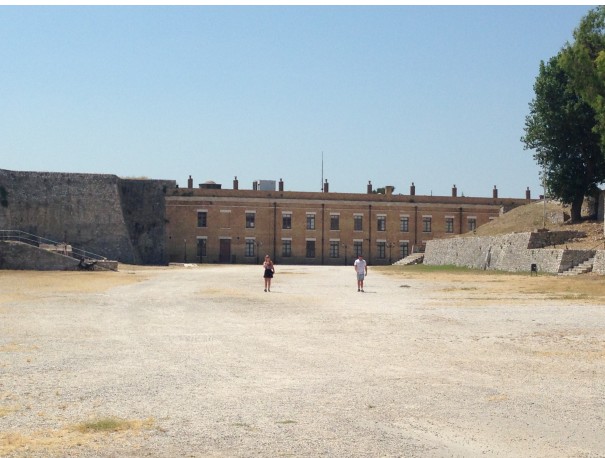

**Figure 3.** The Old Fortress of Corfu (west view).

According to design guidelines found in the literature, augmented reality locations must be carefully sourced in order to not only minimize environmental distractions, such as the noise of a busy urban soundscape, but also support narrative relevance to the represented scenario [38]. Hence, the archaistic and religious character of the selected site surroundings was ideal for adding a layer of conceptual connection with the game's story line and thus enhance the immersion of the augmented environment. Furthermore, virtual audio objects were placed in a way matching the layout of the physical environment [39]: the sound of the Italian sailor was assigned to random points around the near-shore temple, the sound of the Plague monster to dark cavities in the stone wall, and the sound of the naval bombardment along the edge overlooking the sea (Figure 4).

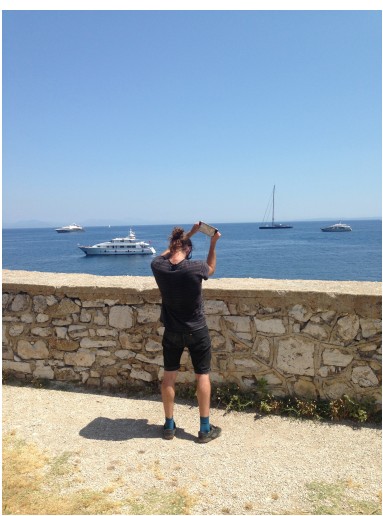

**Figure 4.** The area assigned to naval bombardment.

The experimental sessions were completed within two days and lasted for a total of twenty hours. Two sets of equipment (tablet and headphones) were available allowing for the participation of a maximum two subjects at a time. Each session lasted approximately 30 min with every participant receiving instructions, then playing one, two, or all three game scenarios, and finally completing the respective questionnaires. The experimenters were observing from a distance and provided help when needed. The participants were informed about the purpose of the study, assured that their feedback would be anonymous, and made to feel free to submit it without fear of negative consequences. In the cases of minors, the written consent of their parents was sought. In total, 30 volunteers (16 males and 14 females) were recruited. In terms of their age, they were categorized as follows: 7 were below 18, 8 between 18 and 25, 9 between 26 and 35, and 6 over 35 years old. All seven minors were pupils,

whereas the adults consisted of 10 university students and 13 with a different occupation. The majority of the subjects were fans of technology (29/30) and actively involved with music (20/30) or electronic games (22/30). Around half of the subjects were not accustomed to audio applications (14/30) or had never had previous experience with an AR application (18/30), whereas a very small minority had ever played an AG (4/30) or an ARAG (2/30) before. The diversity of the subjects can be seen as a strength from a methodological perspective for facilitating the investigation of the game experience among distant groups.

The experiment was conducted according to the following procedure: First, the participants (one or two of them at a time) completed the demographic/technographic questionnaire. Next, they were given a brief introduction to the scope of the research and the tales of Saint Spyridon, which accounted for the scenario of the game, and then, they selected a GM with which to start. In case both players would play the same GM at the same time, the respective sounds were assigned to different random positions for each player. After making their selection, the participants were briefly explained how to wear the headphones, hold the device, and perform the necessary gestures. No further time was given to get accustomed to the equipment. After completing one GM, each participant would choose either to try another one or to finish the game. At the end of the sessions, the respective questionnaires were administered to the subjects depending on how many GM in which they participated.

### 4.3. Analyzing the Results

All 28 subjects that played the Famine game completed the objective (100% success rate). The Plague game was also played by 28 subjects with just one failing to complete (96.4%), whereas the Siege game was completed by 16 out of 22 subjects (72.7%). In all GM, there was no player that did not complete the exploration-phase. Additional help was requested by two subjects in the Famine game (7.1%), four in the Plague game (14.2%), and five in the Siege game (22.7%). Almost all help questions referred to confirming the correct positioning of the body and tablet once in the action-phase. Regarding the time taken for completion, a significant randomness of time variations was observed, and it was thus decided to exclude the specific factor from the analysis. Since time was not mentioned in the instructions, each participant managed it in their own way: some walked slowly to enjoy the soundscape, others sped up to finish quickly, while a few others delayed deliberately to see whether the game would crush under any bugs.

According to the GMSQ, all GM were well received by the participants; the Famine game was rated with an average mark of 4.19 out of 5, the Plague game with 4.01, and the Siege game with 3.57. The ratings in the OEQ showed an average mark of 4.16 out of five. When focusing on the ratings per participant, only two averages were below 3.5 (3.25 and 3.33 respectively), while 16 of 30 averages were above 4 with the the highest average at 4.77. Extracting the mean average of those averages revealed a high positive acceptance response of 4.01 for the overall experiment.

The demographic/technographic analysis showed no critical differences among the diverse groups of subjects. An Analysis Of Variance (ANOVA) was performed for each category (*t*-test for binary groups) to compare their response in relation to five average ratings: the three GMSQ, the OEQ, and the mean of player-specific averages. In total, nine categories were investigated: age (4 groups; Figure 5), occupation (3 groups), sex (2 groups; Figure 6), familiarity with music (2 groups), familiarity with audio applications (2 groups), familiarity with electronic games (2 groups), familiarity with audio games (2 groups), familiarity with augmented reality games (2 groups), and familiarity with augmented reality audio games (2 groups). In all cases, the *p*-factor was above the critical threshold ($p > 0.05$).

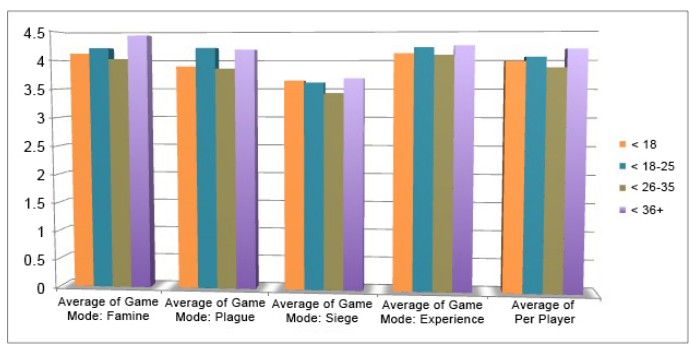

**Figure 5.** ANOVA: Age.

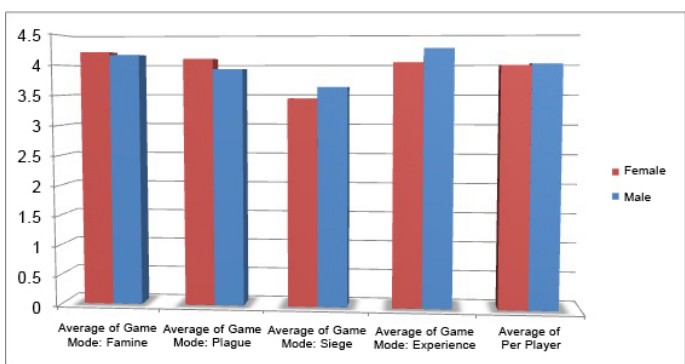

**Figure 6.** ANOVA: Sex.

Regarding the sets of statements targeting usability and immersion in the GMSQ (Figure 7), the averages for the Famine game were 4.29 and 4.04, respectively, for the Plague game 3.94 and 4.11, and for the Siege game 3.43 and 3.72. The similar sets of statements in the OEQ generated an average of 4.35 for usability and 4.2 for immersion. Figure 8 demonstrates the average ratings on the difficulty of each phase. The exploration-phase was rated similarly for all GM: 4.0 for Famine, 3.7 for Plague, and 4.0 for Siege (where 5 corresponds to most easy to use). The action-phase of the Famine game was rated even higher with 4.18, possibly because players were already accustomed to that same mechanic in the exploration-phase. The action-phases of the Plague and the Siege game were rated as 3.41 and 2.81, respectively, suggesting that the more complex the interaction, the greater was the perceived difficulty. An ANOVA analysis was performed to investigate the relation between the perceived difficulty and the users' immersion. Focusing on the Plague and Siege game, which featured more complex gestural interaction in their action-phase than in their exploration-phase, subject groups discerned by their ratings on the difficulty of each phase were compared with respect to their usability and immersion assessment for each GM. Regarding usability, a significant difference ($p < 0.05$) was found in the responses of subjects who reported no difficulty in sonic interaction within the exploration- and action-phases of the Plague game, a finding that hints at the possibility that the subjects accepted the transition to a more complex interaction mode as a level-up in the course of the game. In the Siege game, both extremes, i.e., the highest and lowest ratings on perceived difficulty, were found critical in the action-phase. However, no significant difference was found in any of these cases regarding immersion in the game process ($p > 0.05$), suggesting that the perceived difficulty of complex gestures did not obstruct the users' experience.

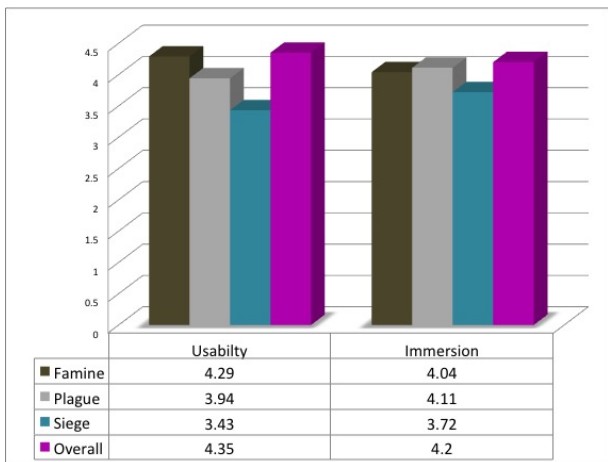

**Figure 7.** Usability and immersion in the GMSQ.

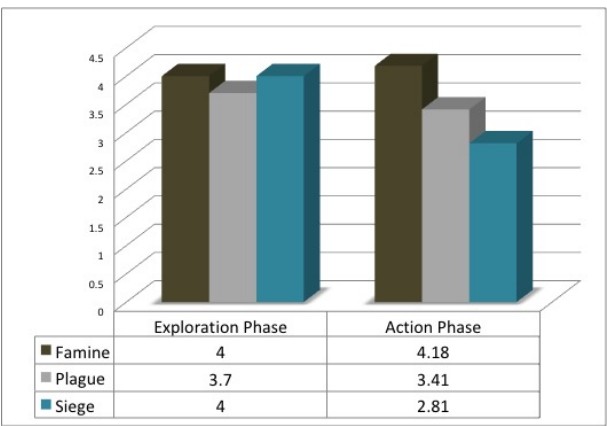

**Figure 8.** Demonstration of the average ratings on the difficulty of each phase.

Regarding response extremes, the lowest average in the Famine game was generated from the responses to the statement "The sounds of the game had no emotional impact on me" (2.82 out of five inverted, i.e., players' emotional response was rather low). All other statements generated an average above 4.0, with the highest one in the statement "It was difficult to complete the game" (4.50 inverted, i.e., the users found the game mode quite easy). In the Plague game, the same statement on emotional impact resulted in the lowest average (3.04) followed by the statement "It was difficult to interact with the sound during the second phase of the game" (3.41), whereas the highest average was generated from the statement "I liked the game" (4.50), followed by "Audio action and interaction through the device were successfully synced" (4.43). In the Siege game, three averages were rated below 3.0, namely statements "It was difficult to complete the game" (2.68), "It was difficult to interact with the sound during the second phase of the game" (2.81), and the "usual suspect" of emotional impact (2.86). An average of four and above was also generated from just three statements, namely "The sounds of the game helped me concentrate on the gaming process" (4.10), "I liked the game" (4.05), and "It was difficult to locate the sound during the first phase of the game" (4.0 inverted).

Further analysis of those findings revealed that, in terms of the lowest averages, the low ratings on the emotional impact were significant to the overall rating of the Famine game ($p < 0.05$), but not for the other two game modes ($p > 0.05$), whereas the perceived high difficulty of the game mode's second phase (action) had a significant effect on the overall rating of the Siege game ($p < 0.05$), but not on the Plague game mode ($p > 0.05$). In terms of the highest averages, in Famine, the perceived ease in completing the game mode was significant to its average rating ($p < 0.05$), whereas in Siege, it was the highest ratings on concentration that showed a significant effect. In Plague, quite a few factors had to be analyzed before a significant interaction could be found, since for the top

averages (general impression, synchronization of sound and gesture, concentration, and fidelity of the augmented soundscape), there were hardly any "negative" ratings. The first factors found to have a significant impact on the overall positive evaluation were the quality of sound design and the perceptual clarity of audio positioning.

In the OEQ evaluation case, the overall experience was found strongly positive with an average score of 4.23. The statement "Participating in the game process enhanced my understanding of information related to the cultural heritage of Corfu" was rated with an average value of 3.52. The headset and tablet were found not to be tiring to use (the average score values were equal to 4.40 and 4.0, respectively). Participants rated the statement "During the game I felt that my acoustic ability was enhanced" with an average score of 4.03 and manifested their willingness to participate again in an ARAG experience with a high average score of 4.80, the highest average among all ratings.

The free-form comments that 17 subjects submitted were in general very positive. The game was characterized as a "very nice", "special", "interesting", "different", and "awesome" experience, a "Hidden-Treasure game with sound". The subjects also shared a wide range of feelings: in the Famine game, the Italian speaking was "really annoying", and it was "stressing that he was always staying behind"; the Siege game was "intense", while the Plague game was found "frightening" and "scary" by four subjects and "like a fairy tale" by another one. The main problem derived from the provided set of comments was the difficulty of the action-phase in the Siege game: five subjects complained about the lack of feedback on aiming the cannon ball. Another issue highlighted was related to the environmental conditions: the wind was blocking careful listening, and the summer heat was discouraging from prolonged play. Two subjects would prefer a smaller device, and two others pre-recorded instructions. Last, two subjects thought that the field of interaction in the action-phase should be 360°.

*4.4. Discussion*

The research described above evolved around one game with three different game modes and involved the participation of 30 subjects. No control group was used to provide a benchmark that would help measure not only any negative, but also any potential positive impact. Bearing in mind the limitations of our sample size, this research can be seen as a preliminary study on the expansion of sonic interaction within an ARAG environment through gestural activity with the handheld device and needs to be further extended for extracting generalized conclusions on this emerging genre. However, the results regarding the RQs posed can be characterized as promising. All subjects completed the audio navigation task, an observation that is aligned with existing studies on auditory way-finding aid, and quite a few succeeded in the gestural sonic interaction task, as well. In all GM cases, the overall experience and the experience per player were highly rated by the subjects, who additionally expressed great interest in participating in future implementations. This positive response was seen in all subject groups with no significant differences in terms of background parameters, a finding that suggests that the ARAG genre may equally address a broad and diverse audience.

Usability and immersion, the two main considerations defined for this research, achieved high scores in all game modes and for the overall experience as well. Thus, our preliminary research suggests that gestural sonic interaction is both feasible and effective within the generic ARAG context. Players perceived that the difficulty level was increased as the interaction complexity was rising, but provided that (a) no time for training was allowed, (b) the increased difficulty level was found significant to the usability rating of only one GM without affecting its immersiveness, and (c) the overall usability and immersion ratings were high, it can be concluded that difficulty was received not as a negative determinant, but as a positive contributor to a challenging game experience. Furthermore, other factors, such as sound design and fidelity of the game world in terms of the positioning, movement, and synchronization of sonic events, were also found to affect immersion. Nevertheless, it must be noted that many subjects requested more in-game guidance as the interaction mode became more complex. One possible explanation for that is that players were willing

to try challenging interaction modes, but without carefully designed feedback, they may experience frustration. Another explanation draws on existing findings that humans localize the azimuth of a sound source much better than elevation [58], which was more fundamental to the sonic behavior in the action-phase of the Siege game than of the Plague game. In any case, as ARAGs benefit from more complex interaction modes, the threshold distinguishing a challenging from a frustrating experience must be further investigated. Some preliminary guidelines towards that direction, mostly drawn empirically from the experimental sessions, would be to provide players with a thorough tutorial, to select the game location with consideration of environmental conditions, to attribute sounds with clear spatialization properties including position and movement, and to avoid any inconsistencies when designing the users' field of action.

Regarding other issues, the unexpected low ratings on the emotional impact contradict the high ratings on immersion. It turned out that the specific statement was poorly understood, stressing the need for more careful formulation. Many subjects aurally stated after the experiment that they thought only a positive emotional effect was meant. This may explain to some extent why they had rated the specific statement poorly, before leaving a comment that, for example, the Italian irritated them. Last, the statement about the communication of cultural heritage just scratched the surface of the ARAG educational potential. It was not poorly rated, but not as highly as expected either. One possible explanation is that the simple description of the game's scenario without any thorough explanation or in-game narration was insufficient to serve this particular educational scope.

## 5. Conclusions and Future Directions

In this research, the ARAG "Audio Legends" was designed, implemented, and tested, aiming to evaluate the utilization of sonic interaction in an AR environment. Three interaction modes were investigated: audio navigation to a sound source and gestural interaction by aiming and waving the device to hit a sonic target, and by calculating the sound's trajectory and holding the device to block it. The experimental results obtained suggested that these mechanics can successfully express a challenging game experience that is understood and enjoyed by diverse audience groups. Increased interaction complexity results in increased game difficulty, but this was not found to affect negatively the system's usability and players' immersion. Other factors, such as sound design, sound spatialization, and sound-gesture synchronization, as well as environmental conditions are also vitally important to the immersion in the game world.

ARAGs, resulting from the fusion of AG and AR practices, are a new and yet-to-be explored medium that can be systematically applied in research and entertainment. This paper aimed to contribute towards this direction by providing findings and inspiration to prospective ARAG developers. The authors' future considerations include raising public awareness about the genre through a variety of game scenarios, enriching the results by investigating further modes of sonic interaction, and exploring the genre's educational potential in targeted instructive activities. Bringing people who have never had any such prior experience in contact with the ARAG genre may not only result in a novel popular medium, but also enlarge the sample size for similar future research and thus provide accurate insight and allow for the systematic study of the field.

**Author Contributions:** Conceptualization, E.R. and A.F.; methodology, E.R. and A.F.; software, K.V.; validation, E.R., K.V. and A.F.; formal analysis, N.M.; investigation, N.M. and A.F.; resources, E.R. and N.M.; data curation, E.R. and K.V.; writing, original draft preparation, E.R.; writing, review and editing, A.F.; visualization, E.R.; supervision, A.F.; project administration, A.F.; funding acquisition, A.F.

**Funding:** This research was funded by Operational Program Human Resources Development, Education and Life-long learning, Priority Axes 6, 8, 9, Act "Supporting Researchers with emphasis on New Researchers" (grant MIS number: 5007016), which is co-financed by Greece and the European Union (European Social Fund (ESF)).

**Conflicts of Interest:** The authors declare no conflict of interest. The funders had no role in the design of the study; in the collection, analyses, or interpretation of data; in the writing of the manuscript; nor in the decision to publish the results.

**Abbreviations**

The following abbreviations are used in this manuscript:

ARAG    Augmented Reality Audio Games
AG    Audio Game
AR    Augmented Reality
ARA    Augmented Reality Audio
GM    Game Modes
IP    Interaction Phases
RQ    Research Questions
GMSQ    Game Mode Specific Questionnaire
OEQ    Overall Experience Questionnaire
ANOVA    Analysis Of Variance

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
