# Peer review of "Audio Legends: Investigating Sonic Interaction in an Augmented Reality Audio Game"

_mti, doi:10.3390/mti3040073_

Round 1

Reviewer 1 Report

The topic is highly interesting. There has not been much research yet in this upcoming field, and there is much need for the development of evidence-based best practices, both for audio-only and audio-augmented AR applications. This research could be a step towards developing those.

The main problem I see with this research is the self-developed questionnaires. The questions are not well thought through and do not consider the definitions of the aspects of interest. This is difficult to "fix" after the fact. Still, I think it is important more research like this is published. Providing a little more analysis and being a little more careful with the conclusions that are drawn could address some of the issues. Below some more detailed notes/tips per chapter.

The background chapter (Ch2) is thorough on audio games and augmented reality audio. 

Methods: the main weakness here seems to be the use of self-developed questionnaires without much reference to definitions or aspects of game experience and immersion as defined in existing literature. I understand the need to ask about specific aspects present in this particular game, but now you have issues considering formulation (about "emotion" mentioned in discussion, but other questions also contain jargon and/or seem open to multiple interpretations, e.g. "I found the sound design of the game satisfying") and many immersion questions are not about immersion specifically. Actually, the immersion questions in the overall experience questionnaire do not seem to be about one aspect at all, but a variety of aspects (acoustic ability, game experience, educational, replay).

It would also have been better to include an semi-open interview at the end, so the participants would easily be able to provide more qualitative input. The authors do mention some participants' comments, but then you are dependent on whether participants provide this input themselves. Through an interview you could obtain input on specific topics, also from participants who are less talkative.

In the discussion, the authors mention the game prototype was highly accepted. I think this is a difficult statement to make, considering it is based on a self-made questionnaire for which no statistical data is available from other evaluations to use for comparison.

In the conclusions, it is stated that that the "increased interaction complexity [as a result from the gestures] has not been found to negatively affect the system's usability and players' immersion". I don't see the analysis to support this in the results section. Perhaps an addition of a graph like Figure 7 but then comparing Exploration and Action (so navigation vs gesture interaction for Plague and Siege) on Immersion and Usability, could provide the necessary evidence?

Finally, I think that it would be very useful for the field in general to provide some preliminary ARA interaction guidelines based on your experiences so far, possibly with the note that their robustness could be investigated in follow-up research.

Author Response

Dear Reviewer,

we would like to thank you for the prompt reviewing process of our paper entitled “Audio Legends: Investigating Sonic Interaction in an Augmented Reality Audio Game”. Your comments were accurately described and useful for improving the overall quality of our research presentation. We have resolved all your comments  and we have also implemented all the suggested corrections and necessary changes in the revised manuscript. Please note that in the revised paper version, any text that has been removed appears under a strike-through effect, whereas text that has been added appears in blue color. Below you may find our analytical response regarding each comment and requested change.

You have pointed out that the self-developed questionnaires should have been better thought and formulated. We fully agree with that remark; however, as you have also mentioned, this is something difficult to fix at this point, since the experimental part of our research has already been completed. We tried to compensate for that weakness by modifying our conclusive statements to be less strong and more carefully formulated in many cases including the ones that you suggested. We also followed the remark to add more analysis particularly regarding our statement that "increased interaction complexity [as a result from the gestures] has not been found to negatively affect the system's usability and players' immersion". This addition is found in line 450. We have also added some guidelines from our experience with the project that can be found in line 538. Last, we also agree that an semi-open interview at the end of our experience would have provided more insight and thus we are willing to apply this methodology to our future research.

With best regards,

(and on behalf of the authors)

Emmanouel Rovithis

Reviewer 2 Report

This is an interesting piece of research. The research questions guiding the study are both novel and timely.

The study is well designed and the analysis of the data seems adequate to answer the research questions at hand.

The findings are also interesting especially the lower ratings for emotional impact which the authors say contradict the higher results gained for immersion.

We know that immersion and emotion are linked in visually-based AR contexts but further research might consider if this holds true for ARAGs.

There are some issues I suggest to correct with this paper. The results obtained are preliminary as they emerge from a single study and are undertaken with a small sample size. As such the statements about the impact of the results need to be made less strong.

In the discussion, you open by stating that the game prototype was highly accepted. You say: "From the above results it is clear that the game prototype was highly accepted". I think this statement is too strong. Instead, I suggest that you say that while this was only one study with a small sample size the results for acceptance were quite good. Similarly, the following statement from the discussion section is also too strong "This positive response was seen in all subject groups with no significant differences in terms of background parameters, indicating that the ARAG genre can successfully address a broad and diverse audience". This result is good and indicates that the ARAG genre might address a broad and diverse audience but your sample size is too small to definitively make this statement. Again the statement "This is a strong evidence that gestural sonic interaction is both feasible and effective within the generic ARAG context." is too strong. I suggest instead that you tone down the claim that the evidence is strong and rather say that your preliminary study suggests that this may be the case. In the conclusions section, you say the following: "The experimental results obtained have revealed that these mechanics can successfully express challenging game-experience that is understood and enjoyed by diverse audience groups." I suggest changing this sentence also to say that the results 'suggest' rather than "reveal".

These issues aside I think the paper is very good and the research is quite interesting. As such I recommend that the paper be accepted with minor revisions. These revisions are the four points outlined above.

Author Response

Dear Reviewer,

we would like to thank you for the prompt reviewing process of our paper entitled “Audio Legends: Investigating Sonic Interaction in an Augmented Reality Audio Game”. Your comments were accurately described and useful for improving the overall quality of our research presentation. We have resolved all the comments and we have also implemented all the suggested corrections and necessary changes in the revised manuscript. Please note that in the revised paper version, any text that has been removed appears under a strike-through effect, whereas text that has been added appears in blue color. Below you may find our analytical response regarding each comment and requested change.

You have pointed out the limitations of our small sample size and thus the need to make our statements about the impact of the results less strong. This has been addressed by modifying our conclusive statements to be less strong and more carefully formulated in many cases. We have also complied to all other suggestions you made regarding statements that need to be changed. These can be found in lines 507, 518, 522 and 557.

With best regards,

(and on behalf of the authors)

Emmanouel Rovithis

Reviewer 3 Report

In this paper the authors present "Audio Legends", an augmented reality audio game that incorporates three modes of interaction. A user study was conducted that examined the usability and immersion within the game was conducted by formulating three research questions that were concerned with i) utilization of complex navigational and gestural sonic interaction, ii) sonic interaction and sound design techniques to facilitate immersion and motivate players to explore and interact, and iii) demographic factors and "technographic" data (familiarity with technological and musical elements, and their effect on the enjoyment on an AR audio game).

Overall, the paper is very well structured and well written making it easy to follow and understand. The Background section is sufficient and provides an overview of the research area (e.g., AR audio games, audio games, etc.). The game itself is well described as are the experimental results.

Overall, I don't have any issues with the work the paper describes and in fact,  the paper focuses on an important topic (i.e., augmented reality audio game) that, in my opinion, has not be investigated enough. The majority of AR work focuses on visuals. Similarly, the majority of games focus on visuals too and sound/sound design is often minimized if not ignored altogether.

Some minor comments:

Introduction could be slightly expanded upon by perhaps expanding upon the importance of sound in games and media in general. That being said, this is described in the background section so perhaps moving some text around. This is of course a suggestion and style preference hence I leave it to the authors to keep/ignore. I just find the Introduction section somewhat brief currently. Expanding upon slightly on the ideas presented for future work. It may be useful to briefly elaborate on what is currently presented. Page 2, line46: should "pursuit" be "pursue"? Perhaps just remove "pursuit to" altogether. Page 4, line 135: "that audio" should be "that the audio". Page 4, line 137: Replace "quite a few" with "many". Page 6, line228: "In Exploration-phase" - perhaps replace with "In th Exploration-phase". When describing how many subjects, when specifying between 0 -9, spell out the word. For example, page 10, line 300 "... requested by 2 subjects ..." should be: "... requested by two subjects ...". I personally would also use "participant" as opposed to "subject". Just a style preference though.

Author Response

Dear Reviewer,

we would like to thank you for the prompt reviewing process of our paper entitled “Audio Legends: Investigating Sonic Interaction in an Augmented Reality Audio Game”. Your comments were accurately described and useful for improving the overall quality of our research presentation. We have resolved all the comments and we have also implemented all the suggested corrections and necessary changes in the revised manuscript. Please note that in the revised paper version, any text that has been removed appears under a strike-through effect, whereas text that has been added appears in blue color. Below you may find our analytical response regarding each comment and requested change.

You have suggested expanding slightly on the ideas about our future work, which is now done in line 568. Another suggestion is to move some parts from the Background Section to the preceding Introduction due to the small size of the latter. We agree that the Introduction Section is rather short, yet we think that its role is very specific and its content clearly structured: in the first paragraph all technological agents of relevance to this paper are introduced, including Audio Games, Augmented Reality, Augmented Reality Audio and Augmented Reality Audio Games, whereas the rest of the Section mentions the scope, means, and outcome of the research and describes the structure of the paper. On the other hand, the Background Section follows a path from Audio Games to Augmented Reality Audio Games by describing the characteristics and evolution of each genre and thus we find that removing a part from this Section would interfere with its coherence. All other suggestions you made regarding wording corrections were followed accordingly and can be found in lines 45, 138, 140 and 248, including replacing numbers with letters when expressing subjects (below 10). Finally, we have included an illustration of the gestures employed in new Fig. 1 as you requested.

With best regards,

(and on behalf of the authors)

Emmanouel Rovithis

Reviewer 4 Report

The paper presents the development details and usability/UX testing results of an augmented reality audio game (ARAG). The abstract and introduction sections are clearly and concisely written. The key assertion made by the authors, that gestural interactivity will enhance immersion via more active participation, is made clear.

Section 2 begins by asserting the value of audio as a supplementary output modality, complimenting graphics to achieve various effects (including increased perceptual depth of virtual environments, ease of navigation within complex spaces and better knowledge retention). This section is well-supported with relevant literature and I found the overarching argument to be convincing; though the range of examples is perhaps a little too broad and towards the end, the section does start to feel like a ‘sound is good!’ checklist. One particular assertion that raised a red flag was line 61-62 “Immersion in the virtual environment is also enhanced when graphics are turned off”. Whilst this is supported with a citation, the source literature documents this finding in what appears to be a strictly 2D, highly abstract (basic geometric shape) graphic environment context. As such, I wouldn’t agree with any statement that extrapolated the idea of immersion being greater without graphics into a general context. The following point on lines 62-63 somewhat adds context but I think this issue still needs clarification. The subsequent paragraph which evidences the value of audio-centric gaming and is a little too descriptive but arguably still relevant.  

Section 2.2 considers various approaches to sound within augmented reality. Here the authors reiterate their observation of sound as a means of supporting navigation and immersion. They also make the assertion that sound design in AR is a largely under-researched area of study. This argument is clearly made and is one that I would agree with. The section continues with a review of various existing ARAG titles that utilise sound for various functions that the authors group into three classes: exploration, navigation and entertainment. This distinction is somewhat helpful but there is arguably some overlap between the three. Some explanation of the differences between the classes, particularly exploration and navigation, would be helpful.

As Section 2 comes to a close, the only significant issue I have is with the support for the authors’ main hypothesis; that gestures can enhance ARAG. The literature referenced in this section doesn’t appear to fully justify the authors’ belief in their hypothesis. Although this hypothesis is seemingly very logical, I think that this point needs reinforcing with exiting literature to clearly make a case for integrating gestural interaction.

Section 3 outlines the details of the ‘Audio Legends’ prototype VRAG. As with previous sections, the writing is very clear and largely presents the design concepts very well. An illustration of the gestural interactions (particularly the hitting gesture) would be very helpful. Technical details in section 3.2 are concise and clear, they also addressed a previous question I had regarding the scale/type of physical environment, as these details are explicitly stated within this section.  

Section 4 documents the evaluation procedure. Research questions are clear and relevant though I felt RQ1 was asking two questions at once and it may be clearer to split up technical feasibility and satisfactory experience. Documenting the full questionnaire is very helpful. Based on the methodology details, I am somewhat concerned that the power of the study to draw meaningful conclusions is limited. The absence of any control group denies us a benchmark for the results and the only means that I can see for comparative analysis would be correlating technical scores with immersion scores (a better-designed audio game increases immersion/enjoyment) which, by itself would be a rather limited conclusion. Without re-doing the experiment, could the data be used to check for differences in immersion/usability between gesture and non-gesture sections of gameplay? If you have already done this, can it be made a bit clearer as I would argue it to be the most meaningful result?

Otherwise, the analysis of demographic and technographic data as a means of eliminating potential confounding variables is certainly helpful. Though not my expertise, data analysis approaches appear well-chosen and the results are clearly and appropriately presented.

The discussion highlights the largely homogeneous data set drawn from a diverse group of participants. Whilst qualification of diversity is rather subjective, based on the range of factors the authors outline on page 10, I would agree that this is a notable and fair observation. Subsequent conclusions regarding points such as high acceptance and positive user experience results are fair and supported by the data.   

Overall I would argue that the work scores quite highly for novelty as the use of audio-only games and gestural interaction within AR are relatively rare by themselves so bringing them together is arguably an original approach. As noted above, the quality of the text is consistently strong with only minor errors noted. Illustrations could use a little editing to improve clarity so I would place presentation at average/high overall. The finer details of the methodology are generally appropriate though a stronger justification of the approach would have improved things further. Being a niche research/design area I feel that reader interest will be high but for a smaller demographic. More extrapolation of the findings to more common audio-visual AR could improve this. I did have a problem with the significance of the content, largely due to the limitations of the study design (absence of a control, no clear baseline, largely subjective results). The extent to which this is a major revision issue is borderline and I have added details for one potential means of improving this issue without re-doing the experiment. Otherwise I would argue the paper should be accepted following what would hopefully be minor revisions.

Summary of recommended edits/additions:

The assertion on 61-62 is a bit contentious and needs further support or clarification of context. Distinction between exploration and navigation could be clearer. Line 132-133 reads like a run-on sentence. Reinforce the case for integrating gestural interaction within ARAG in Section 2 (justify the hypothesis). The abbreviation for Interaction phases can probably be dropped. It’s only used in abbreviated form once and IP is a slightly confusing abbreviation in English, commonly denoting ‘Intellectual Property’. Split RQ1 into two? In the questionnaire section, make the fact that ‘technical aspect’ and ‘usability’ statements are the same thing. Consider additional data analysis, such as comparing gestural to non-gestural sections for usability and immersion scores. Can figures 4 and 5 be enlarged so that the text is clearer?

Author Response

Dear Reviewer,

we would like to thank you for the prompt reviewing process of our paper entitled “Audio Legends: Investigating Sonic Interaction in an Augmented Reality Audio Game”. Your comments were accurately described and useful for improving the overall quality of our research presentation. We have resolved all the comments and we have also implemented all the suggested corrections and necessary changes in the revised manuscript. Please note that in the revised paper version, any text that has been removed appears under a strike-through effect, whereas text that has been added appears in blue color. Below you may find our analytical response regarding each comment and requested change.

You have suggested that due to an over-generalisation one of the references in the background section needs to be clarified; thus, we modified our formulation accordingly in line 61. We also added some more explanation regarding the categorisation of ARAG projects in terms of their scope into ‘navigation’, ‘exploration’, and ‘entertainment’ oriented. This addition can be found in line 154. Another important remark was that our hypothesis, and essentially the drive behind this research (that gestures can enhance the ARAG experience), needs more support from existing literature. We feel that the additions in line 209 improve the justification of our belief. Another weakness pointed out, namely the absence of a control group, is now mentioned in line 509 and adds to the general modification of reducing the strength of our conclusions. We also added an illustration of the gestural interaction and enlarged the text of our illustrations, as suggested. Additional data analysis regarding the comparison between non-gestural and gestural interaction in terms of usability and immersion (in essence this refers to simple gestures versus complex gestures) can be found in line 450 in the context of our argumentation that perceived difficulty of interaction does not obstruct the immersion in the ARAG environment. Last, the abbreviation of Interaction Phases (IP) was dropped as suggested, and a run-on sentence in line 132 was corrected.

With best regards,

(and on behalf of the authors)

Emmanouel Rovithis